# A Retrospective Narrative Mini-Review Regarding the Seminal Microbiota in Infertile Male

**DOI:** 10.3390/medicina58081067

**Published:** 2022-08-07

**Authors:** Bogdan Doroftei, Ovidiu-Dumitru Ilie, Ana-Maria Dabuleanu, Delia Hutanu, Constantin-Cristian Vaduva

**Affiliations:** 1Faculty of Medicine, University of Medicine and Pharmacy “Grigore T. Popa”, University Street, No. 16, 700115 Iasi, Romania; 2Clinical Hospital of Obstetrics and Gynecology “Cuza Voda”, Cuza Voda Street, No. 34, 700038 Iasi, Romania; 3Origyn Fertility Center, Palace Street, No. 3C, 700032 Iasi, Romania; 4Department of Biology, Faculty of Biology, “Alexandru Ioan Cuza” University, Carol I Avenue, No. 20A, 700505 Iasi, Romania; 5Department of Biology, Faculty of Chemistry-Biology-Geography, West University of Timisoara, Vasile Parvan Avenue, No. 4, 300115 Timisoara, Romania; 6Department of Mother and Child Medicine, Faculty of Medicine, University of Medicine and Pharmacy, Petru Rares Street, No. 2, 200349 Craiova, Romania; 7Department of Obstetrics and Gynecology, Clinical Hospital Filantropia, Filantropia Street, No. 1, 200143 Craiova, Romania; 8Department of Infertility and IVF, HitMed Medical Center, Stefan cel Mare Street, No. 23-23A, 200130 Craiova, Romania

**Keywords:** semen, sperm, seminal fluid, spermatozoa, microflora, male infertility

## Abstract

*Background*: Infertility is a global burden that affects both sexes with the male component remaining as an explored yet crucial research field that might offer novel evidence. *Material and Methods*: The present narrative mini-review aims to summarize all existing literature regarding the composition of the seminal microflora in infertile men. We performed searches in PubMed/Medline, ISI Web of Knowledge, Scopus, and ScienceDirect between 2018 and 2022 using a combination of keywords. *Results*: A total of *n* = 33 studies met the eligibility criteria and were further considered. From this, *n* = 14 were conducted on human patients, *n* = 3 on zebrafish (*Danio rerio*), *n* = 5 on rats, and *n* = 11 on mice. In twenty-five out of thirty-three papers, the authors sequenced the 16S rRNA; situations occurred where researchers focused on standard laboratory protocols. *Lactobacillus* and *Bifidobacterium* are widely recognized as putative beneficial lactic bacteria. These two entities are capable of restoring the host’s eubiosis to some extent, blocking pathogens’ proliferation and endotoxins, and even alleviating specific patterns encountered in disease(s) (e.g., obesity, type 1 diabetes) due to prolonged exposure to toxicants in adults or from a developmental stage. Over the years, distinct approaches have been perfected, such as the transfer of feces between two species or conventional rudimentary products with proven efficiency. *Conclusions*: The seminal microflora is decisive and able to modulate psychological and physiological responses. Each individual possesses a personalized microbial profile further shaped by exogenous factors, regardless of sex and species.

## 1. Introduction

According to the American Society for Reproductive Medicine (ASRM), infertility designates the couple’s incapacity to conceive, carry, or deliver a baby following twelve months of regular unprotected intercourse [1].

Based on the latest reports and figures concerning the actual prevalence, the overall estimation reaches 13%, up to 15% worldwide. Of the total number, 30–40% of cases are attributable to both sexes, while roughly 20% is solely due to the male component [2,3,4] and for around fifty million couples, which reflect approximately 15%, to genitourinary tract infections [5]. On the other hand, these percentages may be misleading since less than 2% of all known bacteria strains were successfully cultured and identified through conventional protocols [6].

However, numerous factors are responsible for male genital-tract inflammation, including prostatitis and/or epididymitis [7] in the case of acute or chronic infections related to infertility [8]. Thus, a pro-inflammatory cascade may trigger reactions in the chain reflected by low sperm quality via distinct mechanisms such as exacerbated oxidative stress (OS), hampered accessory gland secretion, anatomical sperm tract obstruction, or direct attack upon sperm by microorganisms [9].

Almost every constitutive site of the human body is colonized by microscopic entities; it is understandable that microorganisms may systematically impair the optimal parameters. Even their simple presence in semen samples may compromise the quality of sperm, with those responsible for contaminations originating from the urinary tract of infected patients during sexual intercourse [10].

Thus, in vitro studies brought insight into how bacteria affected sperm function without implied reactive oxygen species (ROS) or inflammatory cytokines [11,12], but with participation in agglutination of motile sperm, apoptosis, generation of immobilization factors, disturbance of acrosome reaction, and DNA fragmentation [13,14,15,16,17,18,19].

Noteworthy is the topic involving antibiotics regime, which caused controversy on whether pathogens are responsible for semen parameters abnormalities in vivo and how treatment presumably leads to an improvement [20]. For example, *Escherichia coli* is the most isolated microorganism identified and is known to adhere directly to the spermatozoa or synthesizing agents that consequently influence the reproductive potential [21,22].

These aspects discussed above fuel the interest to deepen this spectrum as a branch of research with substantial potential. Therefore, the present narrative mini-review aims to bring together the latest research by offering an updated overview, focusing on the underexplored direction of experimental models coupled with the limited knowledge and experience reported on human patients.

## 2. Methodology

This retrospective narrative mini-review follows the standard procedures established by Green et al. [23].

### 2.1. Database Search Strategy

The database used until inception (April 2022) was PubMed/Medline, ISI Web of Knowledge, Scopus, and ScienceDirect. The combination of keywords contains “microflora” and/or “microbiota” associated with “male infertility” and “semen”, “sperm” “seminal fluid”, and “spermatozoa”.

The adopted PubMed string was: male microbiota[Title/Abstract] OR male microflo-ra[Title/Abstract] OR male infertility[Title/Abstract] AND semen[Title/Abstract] AND sperm[Title/Abstract] AND seminal fluid[Title/Abstract] AND spermatozoa[Title/Abstract].

### 2.2. Inclusion Criteria

Only articles written in English that enrolled human patients and experimental models (mice, rats, and zebrafish (*Danio rerio*)), respectively, conducted between 2018 and 2022 were considered.

### 2.3. Exclusion Criteria

Case report(s)/series, meta-analyses, review(s), standard or systematic, letters to the editor, conference posters, work protocols, and preprints were not considered suitable.

### 2.4. Study Selection

Four independent authors (B.D., O.-D.I., A.-M.D., and D.H.) screened the titles and abstracts of the retrieved results. We completed the assignment of relevant manuscripts based on title, abstract, and full content. Any discrepancy was solved by consent with the remaining author (C.-C.V.).

### 2.5. Limitations of the Study

We concentrated on performing a narrative mini-review rather than a quantitative meta-analysis. This decision derives from data scarcity on this topic.

## 3. Results

A total of *n* = 1127 entries were returned during the established interval, from which *n* = 320 studies were conducted on human patients, *n* = 44 on zebrafish (*Danio rerio*), *n* = 431 on mice, and *n* = 332 on rats.

After we completed the assignment of all studies that initially met the eligibility criteria, we created a time series (2018–2022) in Microsoft Excel^®^ that contains the articles per year of publication, number, and database searched. Except for PubMed/Medline where we applied the aforementioned strategy, we restricted the searches to strictly research articles in English for ISI Web of Knowledge, Scopus, and ScienceDirect. Subsequently, we took each article and removed duplicates and foreign articles, letting only those that had as their main objective to demonstrate microbial translocations in the semen/seminal fluid/spermatozoa/sperm of humans, mice, rats, and zebrafish (*Danio rerio*).

According to the database search and the combination of keywords employed, the results are the following: *n* = 69 (male + microbiota + semen + infertility), *n* = 30 (male + microflora + semen + infertility), *n* = 91 (male + microbiota + sperm + infertility), *n* = 34 (male + microflora + sperm + infertility), *n* = 17 (male + microbiota + seminal fluid + infertility), *n* = 12 (male + microflora + seminal fluid + infertility), *n* = 44 (male + microbiota + spermatozoa + infertility), and *n* = 23 (male + microflora + spermatozoa + infertility). The same technique was applied for the experimental models as well with the mention that “male” was replaced with “zebrafish”, “*Danio rerio*”, “mice”, and “rats”, while “infertility” was removed from the search to increase the chances of covering as many results as possible. Therefore, the situation per each combination of keywords was as follows: *n* = 5 (*Danio rerio* + microbiota + semen), *n* = 12 (*Danio rerio* + microbiota + sperm), *n* = 0 (*Danio rerio* + microbiota + seminal fluid), *n* = 7 (*Danio rerio* + microbiota + spermatozoa), *n* = 4 (zebrafish + microflora + semen), *n* = 8 (zebrafish + microflora + sperm), *n* = 3 (zebrafish + microflora + seminal fluid), and *n* = 5 (zebrafish + microflora + spermatozoa). For rodent models, we found the following: *n* = 93 (mice + microbiota + semen), *n* = 156 (mice + microbiota + sperm), *n* = 29 (mice + microbiota + seminal fluid), *n* = 49 (mice + microbiota + spermatozoa), *n* = 32 (mice + microflora + semen), *n* = 47 (mice + microflora + sperm), *n* = 7 (mice + microflora + seminal fluid), and *n* = 18 (mice + microflora + spermatozoa) on mice, and *n* = 91 (rats + microbiota + semen), *n* = 88 (rats + microbiota + sperm), *n* = 17 (rats + microbiota + seminal fluid), *n* = 33 (rats + microbiota + spermatozoa), *n* = 42 (rats + microflora + semen), *n* = 35 (rats + microflora + sperm), *n* = 9 (rats + microflora + seminal fluid), and *n* = 17 (rats + microflora + spermatozoa). Chronologically, the number of studies published per year is the following: *n* = 37 in 2018, *n* = 45 in 2019, *n* = 75 in 2020, *n* = 104 in 2021, and *n* = 59 in 2022 in human patients; *n* = 1 in 2018, *n* = 6 in 2019, *n* = 5 in 2020, *n* = 21 in 2021, and *n* = 11 in 2022 in zebrafish (*Danio rerio*); *n* = 25 in 2018, *n* = 71 in 2019, *n* = 92 in 2020, *n* = 124 in 2021, and *n* = 119 in 2022 in mice; *n* = 40 in 2018, *n* = 44 in 2019, *n* = 56 in 2020, *n* = 113 in 2021, and *n* = 79 in 2022 in rats. According to the database and subjects (humans, zebrafish, mice, and rats), the number of studies published in this context were: *n* = 40, *n* = 3, *n* = 41, and *n* = 19 in PubMed/Medline; *n* = 34, *n* = 3, *n* = 27, and *n* = 18 in ISI Web of Knowledge; *n* = 84, *n* = 4, *n* = 28, and *n* = 15 in Scopus; and *n* = 162, *n* = 34, *n* = 335, and *n* = 280 in ScienceDirect, respectively. After removing duplicates and studies written in foreign languages, *n* = 33 were considered further. From these studies, *n* = 14 were conducted on human patients, *n* = 3 on zebrafish (*Danio rerio*), *n* = 5 on rats, and *n* = 11 on mice.

### 3.1. Seminal Microflora Analysis of the 16S rRNA through Next-Generation Sequencing

It has been revealed by Monteiro et al. [11] that among all semen samples analyzed, *Enterococcus* was dominant to the detriment of *Lactobacillus*, a scenario that represented 0.5% of all entities discovered. Moreover, *Neisseria*, *Pseudomonas*, and *Klebsiella* increment coupled with lactic acid bacteria depletion is related to oligoasthenoteratozoospermia and seminal hyperviscosity. The discoveries of Alfano et al. [24] are of interest, demonstrating the increase in DNA bacterial amount in contrast with the spermatogenesis of other examined group individuals. *Actinobacteria* and *Firmicutes* were predominant, but the authors highlighted a decrease in richness and diversity in patients with complete germline cell aplasia. This evidence reflects the predominance of *Actinobacteria* and the absence of *Clostridia*. This might further indicate an aging phenomenon of the testes. Testes present tissue-associated symbiotic bacteria-harboring *Actinomycetes*, *Bacteroides*, *Pachybacteria*, and *Proteus* in seminoma normozoospermic men and *Actinomycetes* and *Sclerotinia* in non-obstructive azoospermia. Even though testicular sperm microbiota is low in biomass, it is abundant in contamination (50%-70%). There is significance in genera belonging to *Blautia*, *Cellulosibacter*, *Clostridium XIVa*, *Clostridium XIVb*, *Clostridium XVIII*, *Collinsella*, *Prevotella*, *Prolixibacter*, *Robinsoniella*, and *Wandonia* in immature spermatozoa, according to Molina et al. [25].

The semen within normal parameters contains elevated amounts of *Lactobacillus* and exerts a potent effect on its parameters, accompanied by a balanced ratio of *Prevotella*, *Streptococcus*, and *Staphylococcus* genera [26]. However, there is controversy surrounding this topic. Yang et al. [27] uncovered an abundance of *Lactobacillus* in oligoasthenospermic compared to their counterparts. *Prevotella* was associated with low-quality semen. The most numerous associations in dispermic individuals were *Pseudomonas*, *Prevotella* and *Proteobacteria*, *Firmicutes*, *Actinobacteria*, *Bacteroidetes*, and *Fusobacteria*. *Serratia*, *Acinetobacter*, *Pseudomonas*, *Escherichia*, and *Stenotrophomonas* interfere within current methodologies. These microorganisms systematically impact the semen morphology and deoxyribonucleic acid, even leading to mitochondrial disruption.

Almost every site of the human body is populated. *Proteobacteria* and *Corynebacterium* species are negatively associated with embryos through in vitro fertilization (IVF). Štšepetova et al. [28] considered the presence of *Staphylococcus* species, *Alphaproteobacteria* and *Enterobacteriaceae,* as indicators of embryo and sperm quality. Environmental factors also change the bacterial ratio of body fluids. Yao et al. [29] found that *Staphylococcus*, *Corynebacterium*, and *Corynebacterium_1* genus were present in almost all samples analyzed. The existence of distinct signatures in α- and β-diversity between semen and rectal samples was provided by Lundy et al. [30]. It marks an increase in *Aerococcus* in urine and semen and a decrease in *Collinsella* in the semen of infertile men. The seminal fluid bacterial concentration is lower, but the composition is higher in diversity. There are discrepancies between categories, where azoospermic men have a relative abundance of *Mycoplasma* and *Ureaplasma* compared with *Lactobacillus* in normospermic semen and corresponding vaginal samples. *Gardnerella* is also a part of the microflora in both sexes, whereas *Prevotella* is found solely in women [31]. In Table 1, a summarization of microbiota changes in human patients can be found.

### 3.2. Seminal Microflora Analysis through Standard Laboratory Analyses

Mändar et al. [32] revealed a positive correlation between those with non-sexual and sexual experiences in terms of bacterial concentration and diversity, further dependent on the enrichment of the seminal fluid. Vorobets et al. [33] discovered that *Ureaplasma pervum* and *Ureaplasma urealyticum* caused infections of the genitourinary tract system, identifying *Enterococcus faecalis* known for metabolizing active compounds constitutive of specific drugs [34]. Another marker besides those mentioned above of hormonal impairments after possible fertility dysfunction might be the presence of *Lactobacillus* species in the ejaculate [35]. A possible dysbacteriosis may disturb the integrity of commensal microorganisms, a case when a proliferation of opportunistic pathogens might occur. In this case, even sexually transmitted microorganism balance reported in asymptomatic men could be perturbed, generating a cascade effect on sexual and reproductive health. DNA from *Pseudomonas aeruginosa*, *Chlamydia trachomatis*, *Staphylococcus aureus* and epidermidis, *Klebsiella pneumoniae*, *Neisseria gonorrhoeae*, species of *Lactobacillus,* and *Escherichia coli* were dominant and reported at different values [36].

### 3.3. Seminal Microflora Analysis on Experimental Models

#### 3.3.1. Zebrafish (*Danio rerio*)

Numerous factors influence fertility status, including obesity and chemical compound usage on a large scale. Thus, Su et al. [37] successfully obtained a zebrafish obesity model following egg yolk powder administration. Besides the testicular inflammation, increased pathogenic bacteria proliferation in obese *Danio rerio* was observed. These observations were consistent with the study of Jiang et al. [38], where they tested the single and joint effects of tebuconazole (TEB) and difenoconazole (DIF). A mixture of TEB and DIF displayed additive effects on the acute toxicity but was less pronounced than TEB and DIF alone on the liver, gonad, and intestinal microflora. Valcarce et al. [39] have already shown that a feeding regime based on probiotics may be a time-cost-efficient approach to rescuing low fertility status. The group that received *Lactobacillus rhamnosus* CECT8361 and *Bifidobacterium longum* CECT7347 (1:1) for three weeks had the same weight but improved sperm parameters. In Table 2, a summarization of microbiota changes in *Danio rerio* can be found.

#### 3.3.2. Rats

It has been recently postulated by Wang et al. [40] that Di-(2-ethylhexyl)phthalate (DEHP) toxicity is related to the model used and strains’ changes in the gut. Sprague-Dawley rats were the most vulnerable to bacterial translocations from all four groups: Wistar rats, BALC/C, and C57BL/6J mice. Liu et al. [41] reported an association between gut alteration and defective spermatogenesis, consistent with data obtained by Zhang et al. [42] following the exposure to glyphosate (GLY) and phthalate dibutyl phthalate (DBP). DBP causes seminiferous atrophy and spermatogenic cell apoptosis. A joint effect of fluoride and arsenic reduces testicular weight and hormone levels. Liu et al. [43] showed a positive association with cells’ natural degradation process. The vaginal microbiota probiotic *Lactobacillus crispatus* impacts sperm activity. Li et al. [44] acknowledged that *Lactobacillus crispatus* is responsible for low-number-related pregnancies due to its adhesion property and even accounts for some unexplained infertility. In Table 3, a summarization of microbiota changes in rats can be found.

#### 3.3.3. Mice

A high-fat diet (HFD) for prolonged periods of time ultimately leads to metabolic disorders and impaired sperm production in males [45,46], associated with high circulating endotoxins and decreased spermatogenesis. Conventional dietary products mainly target obese-induced dysbacteriosis [47,48] following the administration of *Lactobacillus fermentum* NCDC 400 and *Lactobacillus rhamnosus* NCDC 610 or type 1 diabetes [49,50] through fecal microbiota transplantation (FMT) [51]. Moreover, it has been demonstrated on previous occasions that gut flora, semen parameters, and testosterone deficiency might be influenced in both ways by probiotics *Lactobacillus plantarum* TW1-1 [52], alginate oligosaccharides [53], or selenium [54] and doxycycline [55]. In Table 4, a summarization of microbiota changes in mice can be found.

### 3.4. Potential Therapeutic Approaches

According to the World Health Organization (WHO), probiotics are living microorganisms derived mainly from cultured dairy products. On the other hand, prebiotics is committed to inducing the growth or activity of beneficial microorganisms. Presently, probiotics are the most powerful and safe alternative in re-establishing the host’s eubiosis with proven efficiency in almost every field of expertise when applied. There is currently one randomized controlled trial underway titled “*Effect of Antioxidant Probiotic Administration on Seminal Quality and Reproductive Outcomes*.” in the recruitment stage and expected for completion in May 2023 (NCT04585984). The estimated number of participants is *n* = 280 but limited to a single center. Briefly, the authors aim to allocate individuals into two equal groups (*n* = 140) and subsequently administer a mixture of *Lactobacillus rhamnosus* and *Bifidobacterium longum* for three weeks.

Retrospectively, Barbonetti et al. [56] discussed their preliminary data according to which probiotics combination of *Lactobacillus brevis* (CD2), *Lactobacillus salivarius* (FV2), and *Lactobacillus plantarum* (FV9) confer protection of human spermatozoa from radical oxygen species in case of a vaginal disease, thereby improving the chances of fertilization potential. Various studies subsequently emphasized the role of probiotics in improving fertilization potential and related endocrine and sperm parameters. As previously indicated by Valcarce et al. [57], a 3-week regime with *Lactobacillus rhamnosus* CECT8361 and *Bifidobacterium longum* CECT7347 significantly enhances sperm quality parameters in asthenozoospermic males, followed by a notable decrease of DNA fragments and intracellular H_2_O_2_. Moreover, during a 6-month diet with *Lactobacillus paracasei* B21060 (Flortec) administration in infertile men, the authors also observed a refinement in the volume of the ejaculate, sperm concentration, progressive motility, and the percentage of typical forms as indicated by Maretti et al. [58]. The same team further noted an improvement in hormone levels compared with the control group after the Flortec regime [58]. The same parameters were investigated, excepting sperm count, live sperm, and serum and seminal total antioxidant capacity. Helli et al. [59] revealed a decrease in plasma pro-inflammatory, serum, and seminal malondialdehyde after a diet containing *Lactobacillus* and *Bifidobacteria* species. This strategy was successfully translated to murine models as well, among the most preferred species to study reproductive potential, including *Lactobacillus coagulans*/*casei*/*rhamnosus* PB01 (DSM 14870), *Lactobacillus* and *Bifidobacterium*, and occasionally *Candida utilis* (Cu. M02) and *Streptococcus thermophilus* (St. S07) [59,60,61,62,63]. *Lactobacillus rhamnosus* CECT8361 and *Bifidobacterium longum* CECT7347 had a beneficial role in humans and rodents, as suggested by Valcarce et al. [39]. Separate lines of evidence proved the same in the case of prebiotics. They increased the *Lactobacillus* and *Bifidobacterium* ratio, thereby elevating short-chain fatty acids essential to regulate metabolic and immune function [64,65]. For example, Rodrigues et al. [66] found that oligofructose supplementation in the diet of mice promoted an alteration in steroidogenesis that subsequently affected plasma corticosterone and testosterone levels.

Moreover, another alternative to alleviate dysbiosis involves the installation of processed stool, obtained from a healthy individual, in a patient’s small intestine. Thus, FMT became a so-called method of choice, where Zhang et al. [67] even demonstrated how FMT leads to male fertility ameliorations. This technique induced significant shifts of beneficial bacteria such as *Bacteroidetes*, *Bifidobacterium*, *Sphingomonas*, and *Campylobacter*. Their presence was associated with the enhancement of spermatogenesis-related genes involving blood and testicular metabolome, spermatogenesis, and even the protein expression of glutathione peroxidase 1 (GPX1). Despite its benefits, the number of papers focused on FMT to treat male infertility in humans is absent from the current literature.

## 4. Conclusions

Compared to the existing evidence in the literature, this manuscript aimed to shed light on an already underexplored field of research found in its infancy stage by providing a different perspective that also involves experimental models. Concerning experiences involving human-related research, certain limitations, legal jurisdictions, and social and ethical values should be imperatively applied and overwhelmed, which is why we wanted to reflect the role of this branch by including mice, rats, and zebrafish (*Danio rerio*) since they are the promotors of a myriad of discoveries translated into the clinical practice.

Probiotics containing *Lactobacillus* and *Bifidobacterium*, which are putative beneficial microorganisms, may restore the eubiosis following the administration of a specific diet to mimic symptomatology. However, referring to the overall potential in countering the associated changes in obesity, type 1 diabetes, and acute toxicity due to prolonged exposure is yet to be explored. Moreover, recent studies contributed to the uprising trend of knowledge. Novel approaches include the transfer of fecal matter between two subjects, how conventional products alleviate the symptoms, and the presence of microorganisms interfere with the pregnancy-related numbers. Starting from our rationale, we will soon contribute to an increasing trend of information regarding how probiotics impact reproductive status since the existing literature supports this kind of relationship.

However, it is hard to argue that distinct urogenital sites harbor a unique microbiome because of the difficulty to acquire exudates free of contamination. One possible explanation resides within the capacity of locally adapted communities for the predisposition of dispersal and colonization via sexual transmission.

Thus, from our point of view, we hope this mini-review may accentuate the scarcity of data and be further used as a support pillar to encourage research teams and increase interest in new large-scale studies that could respond to the plethora of questions regarding how microorganisms impact reproductive traits and success, regardless of sex, and arguably fill this gap in our understanding.

## Figures and Tables

**Table 1 medicina-58-01067-t001:** Summarization of the studies in which the authors sequenced the 16S rRNA in human individuals.

Number of Participants	Hypervariable Region	Sequencer	Microbial Changes	Reference
*n* = 118 participants	V3–V6	Ion PGM	*Proteobacteria* ↑ *Neisseria* ↑ *Klebsiella* ↑ *Pseudomonas* ↑ *Lactobacillus* ↓	[11]
*n* = 15 participants	V3–V5	454-GS Junior	*Actinobacteria* ↓↑ *Bacteroidetes* ↓ *Firmicutes* ↓ *Proteobacteria* ↓ *Clostridia* ↓ *Peptoniphilus asaccharolyticus* ↓	[24]
*n* = 94 participants	V1–V2	MiSeq	*Actinobacteria* ↑ *Bacteroidetes* ↑ *Firmicutes* ↑ *Proteobacteria* ↑	[26]
*n* = 10 participants	V3–V4	MiSeq	*Staphylococcus* ↑ *Corynebacterium* ↑ *Corynebacterium_1* ↑	[29]
*n* = 159 participants	V1–V2	HiSeq 2500	*Ureaplasma* ↑ *Bacteroides* ↑ *Anaerococcus* ↑ *Finegoldia* ↑ *Lactobacillus* ↑ *Acinetobacter lwoffii* ↑	[27]
*n* = 50 couples	V2–V3	454 FLX	*Lactobacillus* ↑ *Alphaproteobacteria* ↑ *Bacteroidetes* ↓ *Alphaproteobacteria* ↓	[28]
*n* = 37 participants	V3–V4	MiSeq	*Aerococcus* ↑ *Anaerococcus* ↓ *Prevotella* ↑ *Pseudomonas* ↑	[30]
*n* = 11 participants	V3–V4	MiSeq	*Blautia* ↑ *Cellulosibacter* ↑ *Clostridium XIVa* ↑ *Clostridium XIVb* ↑ *Clostridium XVIII* ↑ *Collinsella* ↑ *Prevotella* ↑ *Prolixibacter* ↑ *Robinsoniella* ↑ *Wandonia* ↑	[25]
*n* = 36 couples	V4	MiSeq	*Mycoplasma* ↑ *Ureaplasma* ↑ *Lactobacillus* ↑ *Gardnerella* ↑ *Lactobacillus jensenii* ↑ *Faecalibacterium* ↑ *Proteobacteria* ↓ *Prevotella* ↓ *Bacteroides* ↓ *Firmicutes*/*Bacteroidetes* ratio ↓	[31]

↑—increase; ↓—decrease; ↑↓—variations rely on the study design/patient allocations.

**Table 2 medicina-58-01067-t002:** Summarization of the studies in which the authors sequenced the 16S rRNA in zebrafish.

Hypervariable Regions	Sequencer	Microbiota Changes	Reference
V3–V4	Miseq PE300	*Lactobacillus* ↑ *Bifidobacterium* ↓ *Proteobacteria* ↑ *Firmicutes* ↑ *Actinobacteria* ↓ *Escherichia-Shigella* ↑	[37]
V3–V4	Miseq PE300	*Proteobacteria* ↑ *Firmicutes* ↑ *Bacteroidetes* ↑ *Fusobacteria* ↓	[38]

↑—increase; ↓—decrease.

**Table 3 medicina-58-01067-t003:** Summarization of the studies in which the authors sequenced the 16S rRNA in rats.

Model	Hypervariable Regions	Sequencer	Microbiota Changes	Reference
Wistar, Sprague-Dawley	V3–V4	MiSeq PE300	*Proteobacteria* ↑ *Firmicutes* ↑ *Firmicutes*/*Bacteroidetes* ratio ↑ *Oscillospira* ↑ *Peptostreptococcaceae* ↑ *Mycoplasma* ↑ *Roseburia* ↑ *Clostridiaceae* ↑ *Sutterella* ↑ *Clostridiales* ↑ *RF32* ↑ *Christensenellaceae* ↑ *Blautia* ↑ *rc4-4* ↑ *Prevotella* ↓ *Actinomyces* ↑ *Arthrobacter* ↑ *Porphyromonas* ↑ *Bacteroides* ↓	[40]
Sprague-Dawley	V3–V4	MiSeq	*Bacteroides* ↑ *Prevotella_1* ↑	[41]
Sprague-Dawley	V4	HiSeq 2500	*Bacteroidetes* ↑ *Prevotella* ↑ *Prevotella copri* ↑	[42]
Sprague-Dawley	338F–806R primers	-	*SMB53* ↑↓ *p-75-a5* ↑↓ *rc4–4* ↑ *Phascolarctobacterium* ↑ *Veillonella* ↑ *Anaerostipes* ↑ *Desulfovibrio* ↓ *Corynebacterium* ↓ *Trichococcus* ↑ *Lachnobacterium* ↑ *Epulopiscium* ↑ *Allobaculum* ↓	
Sprague-Dawley	V4	HiSeq 2000	-	[44]

↑—increase; ↓—decrease; ↑↓—variations rely on the study design/patient allocations.

**Table 4 medicina-58-01067-t004:** Summarization of the studies in which the authors sequenced the 16S rRNA in mice.

Model	Hypervariable Region	Sequencer	Microbiota Changes	Reference
C57BL/6	V3–V4	HiSeq 4000	*Bacteroidetes* ↓ *Verrucomicrobia* ↓ *Firmicutes* ↑ *Proteobacteria* ↑	[45]
C57BL/6J	V4	MiSeq	*Corynebacterium* ↑ *Rikenellaceae* ↑	[46]
C57BL/6J	V3–V4	HiSeq 6000	*Verrucomicrobiae* ↑ *Gammaproteobacteria* ↑ *Mollicutes* ↑ *Bacteroidia* ↓ *Betaproteobacteria* ↓	[47]
KK-Ay C57BL/6J	V3–V4	MiSeq	*Weissella confusa* ↓ *Clostridium sp. ND2* ↓ *Anaerotruncus colihominis* DSM 17,241 ↓ *[Clostridium] leptum* ↓	[49]
KK-Ay C57BL/6	V3–V4	MiSeq	*Bacteroidales* S24-7 group ↑ *Bifidobacterium* ↑ *Akkermansia* ↑	[50]
CD-1	V3–V4	HiSeqTM 2500	*Lactobacillus* ↑	[51]
C57BL/6	V4	MiSeq	*Bacteroidetes* ↓ *Firmicutes* ↑ *Deferribacteres* ↑	[52]
ICR	V3–V4	HiSeq X Ten	*Lactobacillaceae* ↑ *Desulfovibrionaceae* ↓ *Proteobacteria* ↑ *Bacteroidales* ↑	[53]
BALB/c	V3–V4	MiSeq	*Lachnospiraceae* ↑ *Ruminococcaceae* ↑ *Christensenellaceae* ↑ *Lactobacillus* ↑	[54]
C57BL/6J	V3–V4	MiSeq	*Candidatus Saccharimonas* ↑↓ *Ruminococcus1* ↑↓ *Helicobacter* ↑↓ *Anaeroplasma* ↑↓	[55]

↑—increase; ↓—decrease; ↑↓—variations rely on the study design/patient allocations.

## Data Availability

The datasets used and analyzed during the current study are available from the corresponding author on reasonable request.

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
