# Peer review of "A Retrospective Narrative Mini-Review Regarding the Seminal Microbiota in Infertile Male"

_medicina, 2022, doi:10.3390/medicina58081067_

Round 1

Reviewer 1 Report

The manuscript “A Retrospective Narrative Mini-Review Regarding the Testicular Microbiome in Infertile Male” is aims to offer an updated overview of microbial communities and ratio in the testis of infertile male.  
Unfortunately, the study has serious limitans that I exposed below.

It is not clear what novel information this study provides. As was cited by the authors, information describing the the composition of the testicular microbiome in infertile men, already exists. Neither any justification is provided to support expected strategy for selecting publications. At the same time, the Introduction are excessively on irrelevant aspects as the explanation of epidemiology and about multi-factorial nature of infertility. 

The number of studies used as a basis of the testicular microbiome in infertile men donors is very scarce. Only thirty-three studies were assigned to four species (human (14), zebrafish (3), rats (5) and mice (11), so each group included a extremely reduced number of studies (same conditions?).  It is mentioned that “we performed searches using keyworks and a pre-determined strategy", however, is not given any more information in the ms. Thus, studies "samples" evaluated were not all the studies obtained but those that reached certain minimum values provide by key words. Also, the key words considered for this selection coincides with some response variables. Therefore, the selection of the samples (studies) to be analyzed may resulted in a bias in the Results and in spurious conclusions.
It is stated in the Methodology section that “We employed a strategy for selecting publications”. The authors could indicate the scientific foundation of this?

"A total of n= 1127 entries returned the established interval". The authors mentioned that after removing duplicates (how such level was assessed?) and studies written in foreign languages (the authors did not relate these non-inclusive criteria in the MM section, or the technical foundation).

Author Response

Dear Reviewer #1,

We would like to thank you very much for the positive feedback, interest, and time spent reviewing our manuscript. Per your instructions, we made the respective changes that can be found below:

Comments from the Reviewer: The manuscript “A Retrospective Narrative Mini-Review Regarding the Testicular Microbiome in Infertile Male” is aims to offer an updated overview of microbial communities and ratio in the testis of infertile male. 

Unfortunately, the study has serious limitans that I exposed below.

It is not clear what novel information this study provides. As was cited by the authors, information describing the the composition of the testicular microbiome in infertile men, already exists. Neither any justification is provided to support expected strategy for selecting publications. At the same time, the Introduction are excessively on irrelevant aspects as the explanation of epidemiology and about multi-factorial nature of infertility.

Response: Dear Reviewer, compared with the existing literature in which most of the studies focus on human patients, we wanted to expand this branch of research by including mice, rats, and zebrafish (Danio rerio) since they are the promotors of a plethora of discoveries later translated into the clinical practice. As mentioned in the original version, and revised in the present version, we used several keywords to identify based on title, abstract and content data congruent with our aim. We changed the entire Introduction section with new paragraphs to better reflect the aim of this manuscript.

Introduction: „According to the American Society for Reproductive Medicine (ASRM) directives issued regarding infertility definition, it represents the couple’s incapacity to conceive, carry or deliver a baby following twelve months of regular unprotected intercourse [1].

Based on the latest reports and figures concerning the actual prevalence, the overall estimation reaches 13%, up to 15% worldwide. Of the total number, 30-40% is attributable to both sexes, while roughly 20% is sole to the male component [2–4] and around fifty million couples which reflect approximately 15% to genitourinary tract infections [5]. On the other hand, these percentages may be misleading since less than 2% of all known bacteria strains were successfully cultured and identified through conventional protocols [6].

However, numerous factors are responsible for male genital tract inflammation, including prostatitis and-or epididymitis [7] in case of acute or chronic infections related to infertility [8]. Thus, a pro-inflammatory cascade may trigger reactions in chain reflected by low sperm quality via distinct mechanisms such as exacerbated oxidative stress (OS), hampered accessory gland secretion, anatomical sperm tract obstruction, or direct attack upon sperm by microorganisms [9].

Almost every constitutive site of the human body is colonized by microscopic entities, being understandable that microorganisms may systematically impair the optimal parameters. Even their simple presence in semen samples may compromise the quality of sperm, those responsible for contaminations originating from the urinary tract of infected patients during sexual intercourse [10].

Thus, in vitro studies brought insight into how bacteria affect sperm function without implied reactive oxygen species (ROS) or inflammatory cytokines [11,12], participating in agglutination of motile sperm, apoptosis, generation of immobilization factors, disturbance of acrosome reaction, and DNA fragmentation [13–19].

Noteworthy, the topic involving antibiotics regime caused controversy on whether pathogens are responsible for semen parameters abnormalities in vivo and how treatment presumably leads to an improvement [20]. For example, Escherichia coli is the most isolated microorganism identified known to adhere directly to the spermatozoa or synthesizing agents that consequently influence the reproductive potential[21,22].

Bringing into thought all the aspects discussed above, they fuel the interest to deepen this spectrum as a branch of research with substantial potential. Therefore, the present narrative mini-review aims to bring together the latest research by offering an updated overview, focusing on the underexplored direction of experimental models coupled with the limited knowledge and experience reported on human patients.”

Comments from the Reviewer: The number of studies used as a basis of the testicular microbiome in infertile men donors is very scarce. Only thirty-three studies were assigned to four species (human (14), zebrafish (3), rats (5) and mice (11), so each group included a extremely reduced number of studies (same conditions?).  It is mentioned that “we performed searches using keyworks and a pre-determined strategy", however, is not given any more information in the ms. Thus, studies "samples" evaluated were not all the studies obtained but those that reached certain minimum values provide by key words. Also, the key words considered for this selection coincides with some response variables. Therefore, the selection of the samples (studies) to be analyzed may resulted in a bias in the Results and in spurious conclusions.

It is stated in the Methodology section that “We employed a strategy for selecting publications”. The authors could indicate the scientific foundation of this?

"A total of n= 1127 entries returned the established interval". The authors mentioned that after removing duplicates (how such level was assessed?) and studies written in foreign languages (the authors did not relate these non-inclusive criteria in the MM section, or the technical foundation).

Response: Indeed. This branch of research is very scarce, which is why we found it suitable to conduct a narrative mini-review to gather all existing data from 2018 until 2022 to support further research in this field, to highlight how underexplored this topic actually is, and the substantial potential it possesses. As mentioned above, we used a combination of keywords per each database searched including “microflora” and-or “microbiota” associated with “male infertility” and “semen”, “sperm” “seminal fluid”, and “spermatozoa”. The adopted PubMed string was: (((male microbiota[Title/Abstract] OR male microflora[Title/Abstract] OR male infertility[Title/Abstract]) AND semen[Title/Abstract]) AND sperm[Title/Abstract] AND seminal fluid[Title/Abstract] AND spermatozoa[Title/Abstract]. From this, we created a time series in Microsoft Excel in which we summarized per database and year of publication of the studies. Depending on the database, where possible, we restricted the searches to strictly research articles in English (ISI Web of Knowledge, Scopus, and ScienceDirect), whereas, in PubMed/Medline, we used the strategy mentioned above. Based on the returned results, we manually took each article and removed duplicates and foreign articles, letting only those that had as main objective to demonstrate microbial translocations in semen/seminal fluid/spermatozoa/sperm of humans, mice, rats, and zebrafish.

Kind regards and all the best,

Ovidiu-Dumitru Ilie

Reviewer 2 Report

The title describes testicular microbiome whereas most of the work is related to semen microbiological analysis which is misleading. Of course, most relevant is the microbiome in semen which derives from epididymis, prostate and seminal vesicles as well as from the urthra, but not from the tests. Therefore, the title has to be changed accordingly.  

Author Response

Dear Reviewer #2,

We would like to thank you very much for the positive feedback, interest, and time spent reviewing our manuscript. Per your instructions, we made the respective changes that can be found below:

Comments from the Reviewer: The title describes testicular microbiome whereas most of the work is related to semen microbiological analysis which is misleading. Of course, most relevant is the microbiome in semen which derives from epididymis, prostate and seminal vesicles as well as from the urthra, but not from the tests. Therefore, the title has to be changed accordingly.

Response: Dear Reviewer, thank you very much for your esteem observations. Indeed. The old title version was misleading, which is why we changed it. „A Retrospective Narrative Mini-Review Regarding the Seminal Microbiota in Infertile Male”. Moreover, we also checked once again the entire manuscript for typos or other errors and changed, where necessary, the wrong idea of the testicular microbiome with seminal microflora. We also changed the entire Introduction section with new paragraphs to better reflect the aim of this manuscript.

Kind regards and all the best,

Ovidiu-Dumitru Ilie

Round 2

Reviewer 1 Report

The authors replied to most of the reviewer's questions.

The enormous effort made by the authors to improve the manuscript is noteworthy. However, I still consider that the authors should be more accurate about the novel information this study provides. 

Unfortunately, I cannot see the sentence the authors claim to have added in the ms on my request.

Author Response

Dear Reviewer #1,

We thank you very much for your eminent observations and thoughts regarding our revised version.

We hope we followed your instructions closely and lived up to your expectations to respond appropriately.

Comment from the Reviewer: The authors replied to most of the reviewer's questions.

The enormous effort made by the authors to improve the manuscript is noteworthy. However, I still consider that the authors should be more accurate about the novel information this study provides.

Response: Dear Reviewer, please find several new arguments regarding the novelty of our article (they are also mentioned in the revised version of our manuscript in the Conclusions section). “Compared to the existing evidence in the literature, this manuscript aimed to shed light on an already underexplored field of research found in its infancy stage by providing a different perspective that also involves experimental models. Concerning experiences involving human-related research, certain limitations, legal jurisdictions, and social and ethical should be imperative applied and overwhelmed, which is why we wanted to reflect the role of this branch by including mice, rats, and zebrafish (Danio rerio) since they are the promotors of a myriad of discoveries translated into the clinical practice. However, it is hard to argue that distinct urogenital sites harbor a unique microbiome because of the difficulty to acquire exudates free of contamination. One possible explanation resides within the capacity of locally adapted communities for the predisposition for dispersal and colonization via sexual transmission. Thus, from our point of view, we hope this mini-review may accentuate the scarcity of data, further used as a support pillar to encourage research teams and increase interest toward new large-scale studies that could respond to the plethora of questions regarding how microorganisms impact reproductive traits and success, regardless of the sex and arguably, to fill this gap in our understanding.”

Comment from the Reviewer: Unfortunately, I cannot see the sentence the authors claim to have added in the ms on my request.

Response: Dear Reviewer, please accept our sincere apologies if we forgot to add any sentence from the Response Letter in the manuscript. If this refers to how the study search was done, what we mentioned in the Response Letter can now also be found in the revised version of our article. “After we completed the assignment of all studies that initially met the eligibility criteria, we created a time series (2018-2022) in Microsoft Excel® that contains the articles per year of publication, number, and database searched. Excepting PubMed/Medline where we applied the aforementioned strategy, we restricted the searches to strictly research articles in English for ISI Web of Knowledge, Scopus, and ScienceDirect. Subsequently, we took each article and removed duplicates and foreign articles, letting only those that had as main objective to demonstrate microbial translocations in semen/seminal fluid/spermatozoa/sperm of humans, mice, rats, and zebrafish (Danio rerio).”

If there is still uncertainty about the search strategies, the authors closely followed the directions in Green's study (Green BN, Johnson CD, Adams A. Writing narrative literature reviews for peer-reviewed journals: secrets of the trade. J Chiropr Med 2006;5:101–17.). We have multiple times applied these instructions in previously published papers. The most eloquent example is this review titled: “Preclinical Considerations about Affective Disorders and Pain: A Broadly Intertwined, yet Often Under-Explored, Relationship Having Major Clinical Implications” published in MDPI - Medicina, with the following references: 6(10), 504; https://doi.org/10.3390/medicina56100504. Another example is this recently published article in MDPI – Diagnostics “An Updated Narrative Mini-Review on the Microbiota Changes in Antenatal and Post-Partum Depression” 12(7), 1576; https://doi.org/10.3390/diagnostics12071576.

Kind regards and all the best,

Ovidiu-Dumitru Ilie

Reviewer 2 Report

after revision, the manuscript seems now suitable for publication

Author Response

Dear Reviewer #2,

We thank you very much for your feedback and thoughts regarding our revised version.

Kind regards and all the best,

Ovidiu-Dumitru Ilie